# ARA: Adaptive Rank Allocation for Efficient Large Language Model SVD Compression

## Abstract

In the field of large language model (LLM) compression, singular value decomposition (SVD) is a widely studied and adopted low-rank decomposition technique. Since SVD operates exclusively on linear modules, and these modules in LLMs are separated by nonlinear components, SVD can only be applied independently to each linear module. Under a global compression ratio constraint, determining the appropriate rank for different linear modules becomes a critical problem. Existing approaches, such as heuristic algorithms and mask-based training, have made progress in addressing this challenge. However, these methods still suffer from several limitations: heuristic algorithms explore the solution space within restricted regions, while mask-based training struggles to efficiently capture the relationship between singular value spectra and trainable parameters. More importantly, current methods overlook the key property that the gain function is non-smooth at a compression ratio of 1, which often leads the training process to suboptimal local minima. To address these issues, we propose an Adaptive Rank Allocation (ARA) method. Specifically, (1) ARA introduces a dedicated mask design that enables efficient mapping and updating between retained ranks and trainable parameters; and (2) it employs an additional loss function to guide parameter selection toward globally optimal solutions. Experimental results demonstrate that ARA achieves state-of-the-art performance. On the LLaMA2-7B model with a 80% compression ratio, ARA reduces perplexity on WikiText2 from 8.38 to 6.42 and improves average zero-shot task accuracy by 9.72 percentage points compared with uniform compression. These results highlight the effectiveness of our method for rank allocation in SVD-based LLM compression.

## 1 Introduction

Large language models (LLMs) (Brown et al., 2020; Touvron et al., 2023; Yang et al., 2025) have achieved remarkable success across a wide range of natural language processing tasks. However, their enormous parameter sizes pose significant challenges for storage and computational efficiency (Yuan et al., 2024b; Wan et al., 2024). To alleviate these issues, low-rank decomposition methods (Schotthöfer et al., 2022; Xu et al., 2023; Lin et al., 2025; Huang et al., 2025) have been widely studied for model compression. Among them, Singular Value Decomposition (SVD) stands out due to its theoretical guarantee: by the Eckart–Young theorem (Bhatia, 1997), SVD provides the optimal low-rank approximation under the Frobenius norm. This makes it a representative technique for compressing linear transformations in LLMs. Despite its advantages, SVD can only be applied to linear modules, whereas LLMs consist of numerous nonlinear components such as attention computations and activation functions. Consequently, SVD must be applied independently to each linear module, raising a critical challenge: how to allocate the appropriate rank across modules under a global compression constraint. Prior studies (Yin et al., 2024; Lu et al., 2024; Hu et al., 2025) have shown that different layers contribute unevenly to overall performance, emphasizing the need for systematic rank allocation in SVD-based compression.

Several strategies (Wang et al., 2020; Gao et al., 2024; Yuan et al., 2024a) have been proposed to address this challenge. Heuristic approaches, such as Sensitivity-based Truncation Rank Searching (STRS) (Yuan et al., 2024a), independently evaluate the performance of each module under discrete rank settings(Fig.1(a)) and select rank configurations based on a uniform performance threshold. However, their search space is limited and inter-module dependencies are ignored. Mask-

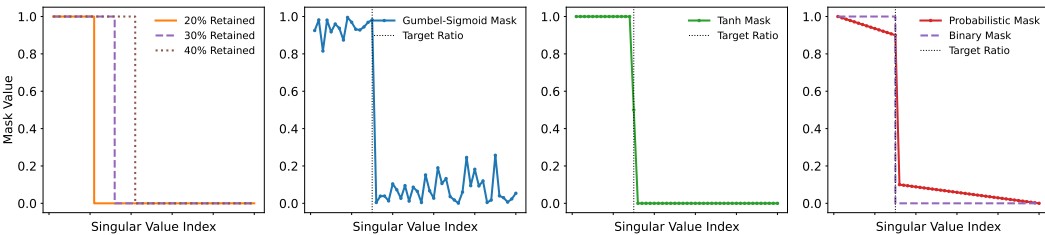

Figure 1: A comparison of our masking strategy against prior work. Prior methods for mask design include (a) heuristic approaches with a limited set of patterns, (b) Gumbel-Sigmoid masks with independent parameters for each value, and (c) tanh-based masks controlled by a single soft cutoff parameter. In contrast, (d) our method utilizes a probabilistic mask to derive a final binary mask.

based methods aim to learn rank selection directly over the singular value spectrum. For instance, ARS (Gao et al., 2024) uses a Gumbel-Sigmoid (Jang et al., 2017) mask, which fails to preserve the same monotonicity as singular values(Fig.1(b)), while Dobi-SVD (Qinsi et al., 2025) employs a $\mathtt{tanh}$-based mask to maintain monotonicity and approximate a binary mask. Yet, the $\mathtt{tanh}$ mask suffers from a local update issue: its parameters are primarily influenced by adjacent singular values, lacking a global perspective(Fig.1(c)).

An even more fundamental issue is often overlooked: the special case where the compression ratio is 1. Given an original weight matrix $W \in \mathbb{R}^{m \times n}$, truncating to rank $k$ produces factor matrices of size $m \times k$ and $k \times n$, with a total parameter count of $k(m + n)$. This decomposed form can exceed the parameter count of the dense representation whenever $k(m+n) > mn$. Consequently, there exists a range of $k$ for which retaining the original dense matrix is strictly more parameter-efficient and can also preserve performance. Nevertheless, current mask-based approaches(Fig.2(a)(b)) either train masks only on pre-truncated low-rank matrices, forcing decomposition on all layers, or train full-rank matrices but do not incorporate the performance advantage of keeping dense matrices during the training stage. Both practices cause the optimization to be trapped in suboptimal solutions.

To overcome these limitations, we propose Adaptive Rank Allocation (ARA), a novel method for allocating ranks in SVD-based LLM compression. ARA introduces a specialized mask module that combines monotonicity preservation with a global perceptual range, addressing the limitations of both Gumbel-Sigmoid and $\mathtt{tanh}$ masks. It also leverages an additional loss function to guide parameter selection toward a global optimum and dynamically handles modules with a compression ratio of 1 by applying trainable masks only when compression is needed(Fig.2(a)(c)). Existing training methods directly use probabilistic masks, while inference employs binary masks; to maintain consistency between training and inference, we adopt the Straight-Through Estimator(STE) (Bengio et al., 2013) to align probabilistic and binary masks(Fig.1(d)) during training.

In summary, our main contributions are as follows: (1) We introduce ARA, an SVD-based rank allocation method, the first to both enable efficient parameter optimization and explicitly consider the necessity of matrix decomposition during training. (2) Extensive experiments on language modeling benchmarks, as well as zero-shot tasks, demonstrate that ARA significantly outperforms state-of-the-art methods.

## 2 RELATED WORK

**SVD for LLM Compression** SVD approximates a weight matrix by the product of two low-rank factors, reducing the number of parameters. Directly applying SVD to weight matrices often leads to performance degradation, motivating several refinements. FWSVD (Hsu et al., 2022) incorporates Fisher information into the decomposition and employs retraining to recover performance. ASVD (Yuan et al., 2024a) scales weight matrices via input-activation normalization, preserving performance at low compression ratios without extensive retraining. SVD-LLM (Wang et al., 2025) derives an activation-aware relationship between singular values and truncation loss, facilitating locally optimal truncation under given constraints. SoLA (Huang et al., 2025) preserves channels dominating MLP activations while applying SVD to the remaining parameters. Dobi-SVD (Qinsi et al., 2025) combines SVD with quantization, allowing more singular values to be retained under

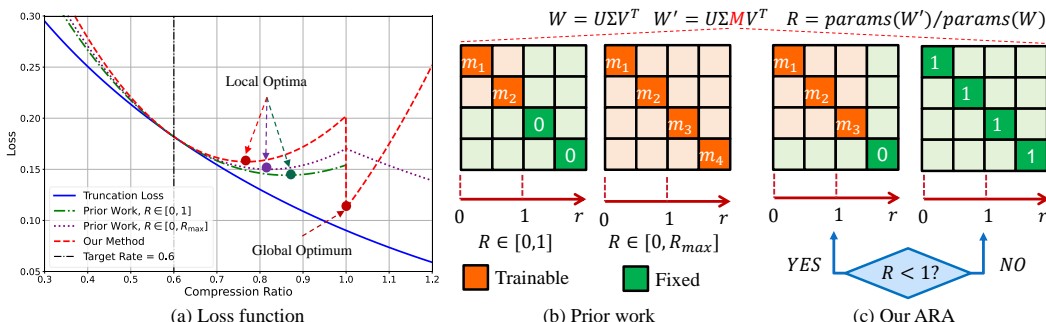

Figure 2: A comparison of our training scope against prior work. (a) Whereas prior work can only find local optima, our method achieves the global optimum. (b) Previous methods operate within fixed low-rank or full-rank training scopes. (c) Our approach dynamically adapts the entire computational flow based on the target compression rate.

the same effective compression budget. These methods primarily focus on final performance, while lacking a systematic study of rank allocation.

**Compression Ratio Allocation** A complementary line of work studies how to allocate compression resources across transformer layers. OWL (Yin et al., 2024) assigns layer-wise compression ratios according to the prevalence of outliers. DLP (Chen et al., 2025) replaces outliers with the median for more stable allocation. AlphaPruning (Lu et al., 2024) and FARMS (Hu et al., 2025) exploit heavy-tailed spectral properties (Barsbey et al., 2021), estimating per-layer training sufficiency and assigning stronger compression to under-trained modules. While based on the model's inherent properties, these methods were designed for pruning (Frantar & Alistarh, 2023; Sun et al., 2024; Ma et al., 2023) and do not exploit SVD-specific structure, limiting fine-grained optimization for linear modules within the same layer.

**SVD Rank Allocation** Heuristic methods such as STRS (Yuan et al., 2024a) evaluate each module under discrete rank settings and select ranks based on a uniform threshold, but this ignores inter-module dependencies and is restricted to discrete states. Mask-based approaches instead learn ranks directly over the singular value spectrum. Wang et al. (2020) trains both the Gumbel-Sigmoid mask and model parameters, while ARS (Gao et al., 2024) reduces cost by training only the mask. However, Gumbel-Sigmoid masks do not guarantee monotonicity with respect to singular value indices, requiring additional alignment losses. Dobi-SVD (Qinsi et al., 2025) addresses this by parameterizing the mask as $m_i = 0.5 \cdot \tanh\left(\beta(k-i)\right) + 0.5$, where $i$ is the singular value index, $k$ a trainable truncation boundary, and $\beta$ a sharpness parameter. This yields near-binary, monotonic masks, but updates are concentrated around $k$, limiting global adjustability. Another critical issue arises when a module's compression ratio exceeds 1: although some existing methods eventually choose to retain the original dense matrix, they do not utilize the full rank during training, preventing the model from correctly assessing the benefit of keeping the dense matrix.

## 3 METHODS

In this section, we introduce our Adaptive Rank Allocation (ARA) method for allocating ranks in SVD-based LLM compression. As illustrated in Figure 3, ARA associates each singular value sequence with a set of trainable parameters, which determine the corresponding probabilistic mask, binary mask, and compression ratio $R$, where $R$ is defined as the ratio between the number of parameters after compression and the number of parameters in the original dense representation. Furthermore, based on the singular value distribution of each module and the current compression ratio, ARA dynamically decides whether to retain the original matrix or to apply low-rank decomposition. This mechanism ensures that when decomposition leads to poor approximation quality, the dense representation is preserved, thereby improving both parameter efficiency and model performance. Under the joint constraints of the cross-entropy loss, the guidance loss, and the compression-ratio discrepancy loss, ARA learns the optimal rank allocation strategy for a specified compression ratio.

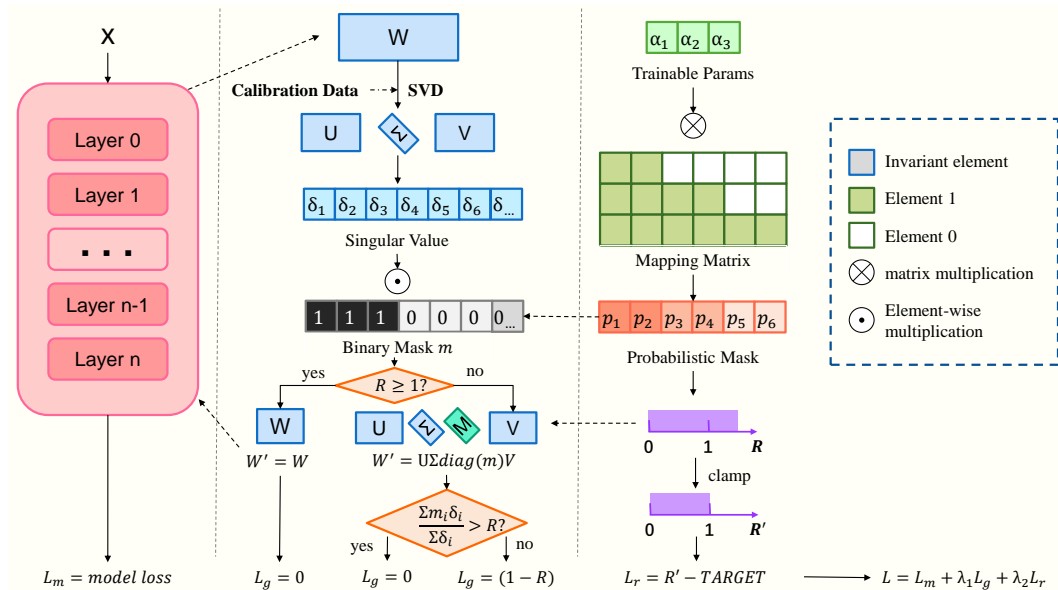

Figure 3: An illustration of our ARA framework. Trainable parameters $\{\alpha_i\}$ produce a probabilistic mask via a mapping matrix, from which a compression rate R and a binary mask M are derived. The rate R dynamically dictates the computational flow and activates a full-rank guidance loss when necessary. The entire model is trained with a joint objective comprising the model loss, the guidance loss, and a compression rate loss.

The following subsections provide detailed explanations of each component, and the pseudocode of ARA is summarized in Appendix A.1.

## 3.1 SINGULAR VALUE DECOMPOSITION

We begin with a brief review of singular value decomposition in the context of linear layers. For a weight matrix $\boldsymbol{W} \in \mathbb{R}^{m \times n}$ with $m \geq n$ and an input $\boldsymbol{X} \in \mathbb{R}^{n \times d}$, the compressed representation is written as $\boldsymbol{W}' = \boldsymbol{W}_u \boldsymbol{W}_v$, where $\boldsymbol{W}_u \in \mathbb{R}^{m \times r}$ and $\boldsymbol{W}_v \in \mathbb{R}^{r \times n}$. The objective is to minimize the reconstruction error under a rank constraint:

$$\min_{\boldsymbol{W}'} \ L_r = \|\boldsymbol{W}\boldsymbol{X} - \boldsymbol{W}'\boldsymbol{X}\|_F \quad \text{s.t.} \quad \text{rank}(\boldsymbol{W}') \leq r. \tag{1}$$

Following Wang et al. (2025), let $\boldsymbol{H} = \boldsymbol{X}\boldsymbol{X}^\top$ and let $\boldsymbol{S}$ be the Cholesky factor of $\boldsymbol{H}$ (so that $\boldsymbol{H} = \boldsymbol{S}\boldsymbol{S}^\top$). Apply SVD to the product $\boldsymbol{W}\boldsymbol{S} = \boldsymbol{U}\boldsymbol{\Sigma}\boldsymbol{V}^\top$, with $\boldsymbol{U} \in \mathbb{R}^{m \times n}$, $\boldsymbol{\Sigma} \in \mathbb{R}^{n \times n}$ and $\boldsymbol{V} \in \mathbb{R}^{n \times n}$. Truncating to rank $r$ yields $\boldsymbol{U}_r = [\boldsymbol{u}_1, \ldots, \boldsymbol{u}_r]$, $\boldsymbol{\Sigma}_r = \text{diag}(\delta_1, \ldots, \delta_r)$ with $\delta_1 \geq \delta_2 \geq \cdots \geq \delta_r$, and $\boldsymbol{V}_r = [\boldsymbol{v}_1, \ldots, \boldsymbol{v}_r]$. Since $\boldsymbol{W}\boldsymbol{S} = \boldsymbol{U}\boldsymbol{\Sigma}\boldsymbol{V}^\top$, we have $\boldsymbol{W} = \boldsymbol{U}\boldsymbol{\Sigma}\boldsymbol{V}^\top\boldsymbol{S}^{-1}$, and hence a natural low-rank factorization is given by $\boldsymbol{W}_u = \boldsymbol{U}_r\sqrt{\boldsymbol{\Sigma}_r}$ and $\boldsymbol{W}_v = \sqrt{\boldsymbol{\Sigma}_r}\,\boldsymbol{V}_r^\top\boldsymbol{S}^{-1}$.

The truncation loss can be written as $L_r = \sqrt{\sum_{i=r+1}^{n} \delta_i^2}$, which shows that preserving larger singular values reduces the approximation error.

## 3.2 MASK GENERATION

ARA generates a mask for each singular value sequence, where the optimal rank is determined by training this mask. The mask is required to satisfy two essential properties: (1) Monotonicity. The mask must share the same non-increasing monotonicity as the singular values. This property is crucial because it ensures that by the end of training, reconstruction error is minimized by preserving the largest singular values(Sec.3.1). Enforcing this shared monotonicity promotes consistency between the model's behavior during training and inference. (2) Global Update Influence. The updates for the mask's parameters should be influenced by the global distribution of singular values, rather

than by local subsets of them. This approach prevents the optimization from converging to a local optimum based on a limited view of the singular value spectrum.

Formally, let $\boldsymbol{\alpha} = (\alpha_1, \alpha_2, \ldots, \alpha_D)$ be a set of trainable parameters constrained to the probability simplex. We define a mapping matrix $\boldsymbol{M} \in \{0, 1\}^{D \times r}$, referred to as a staircase binary matrix. Denote by $v_i$ the number of ones in the $i$-th column of $\boldsymbol{M}$, where $v_i \geq v_{i+1}$. The probability vector $\boldsymbol{p} \in \mathbb{R}^r$ is given by

$$\boldsymbol{p} = \boldsymbol{\alpha} \boldsymbol{M}, \qquad p_i = \sum_{j=D-v_i+1}^{D} \alpha_j. \tag{2}$$

Since $v_i \geq v_{i+1}$ and $\alpha_j \geq 0$, we have $p_i \geq p_{i+1}$, thereby guaranteeing the monotonicity of the mask. Moreover, the gradient $\partial p_i / \partial \alpha_j$ is 1 (if $j \geq D - v_i + 1$), which avoids the vanishing-gradient issue of $\tanh$-based masks. We further set $v_1 = D$ and $v_r = 1$, ensuring that the largest singular value is always preserved while every $\delta_i$ contributes to the training.

Given the probability mask $\boldsymbol{p}$, the compression ratio of a module is computed as

$$R = \frac{(\sum_{i=1}^{r} p_i)(m + n)}{mn}, \tag{3}$$

and the binary mask $\boldsymbol{m} \in \{0, 1\}^r$ is determined by

$$m_i = \begin{cases} 1, & \text{if } i \leq \lfloor R \cdot r \rfloor, \\ 0, & \text{otherwise.} \end{cases} \tag{4}$$

To ensure consistency between training and inference, we adopt the binary mask $\boldsymbol{m}$ during training. Since binarization is non-differentiable with respect to $\boldsymbol{\alpha}$, we employ the Straight-Through Estimator (STE). Specifically, for a loss function $\mathcal{L}$, the gradient is approximated as

$$\frac{\partial \mathcal{L}}{\partial \alpha_j} \approx \sum_{i=1}^{r} \frac{\partial \mathcal{L}}{\partial m_i} \frac{\partial p_i}{\partial \alpha_j}, \tag{5}$$

i.e., the gradients with respect to the probability mask are used as surrogates for those of the binary mask, enabling standard backpropagation.

### 3.3 FULL-RANK GUIDANCE

As discussed in Sec.1, the parameter overhead introduced by SVD implies that when the compression ratio approaches 1, directly using the original weight matrix is more efficient. This introduces a near-discontinuous gain when switching to the full-rank matrix, which can make the loss landscape non-smooth, as shown in Fig.2(a). During low-rank training, the model may fail to capture this critical signal, causing the optimization to remain strictly in the low-rank regime rather than exploring the full-rank option. This indicates that the standard performance loss alone is insufficient, and additional constraints are required.

Recalling the input-aware SVD from Sec.3.1, the truncation loss for compression ratio R is $L_R = \sqrt{\sum_{i=\lfloor R \cdot r + 1 \rfloor}^{n} \delta_i^2}$, while the original model output norm is $L_0 = \|\boldsymbol{W} \boldsymbol{X}\|_F = \sqrt{\sum_{i=1}^{n} \delta_i^2}$. We define a metric $G_R$ to represent the fraction of model capacity preserved at compression ratio R:

$$G_R = \frac{L_0 - L_R}{L_0}. \tag{6}$$

This establishes a direct relationship between the parameter compression ratio $R$ and the preserved capacity of the module, $G_R$. If $G_R > R$, it indicates that the compression preserves a relatively larger fraction of model capacity compared to its parameter cost; thus, compression is preferable. Otherwise, the model should retain the original matrix. Based on this intuition, we define the guidance loss as

$$L_g = \begin{cases} 0, & \text{if } G_R > R, \\ 1 - R, & \text{if } G_R \leq R. \end{cases} \tag{7}$$

The guiding function leverages prior estimation to adaptively encourage certain modules to adopt a compression ratio of 1, thereby retaining the original weight matrix. As a result, the computation

flow is dynamically determined by the compression ratio $R$, which can be expressed in terms of the effective weight matrix $\boldsymbol{W}'$:

$$\boldsymbol{W}' = \begin{cases} \boldsymbol{W}, & \text{if } R \geq 1, \\ \boldsymbol{W}_u \operatorname{diag}(\boldsymbol{m}) \boldsymbol{W}_v, & \text{if } R < 1, \end{cases} \tag{8}$$

where $\boldsymbol{W}_u, \boldsymbol{W}_v$ are the SVD components from Sec.3.1, and $\operatorname{diag}(\boldsymbol{m})$ is a diagonal matrix formed from the binary mask vector $\boldsymbol{m}$. When $R < 1$, this formulation uses the trainable mask $\boldsymbol{m}$ to select the top singular values for the low-rank approximation. Conversely, when $R \geq 1$, ARA utilizes the original full-rank matrix $\boldsymbol{W}$.

Although the trainable mask only takes effect when $R < 1$, ARA still generates masks over a theoretically larger ($R_{\max} > 1$) singular value sequence (Fig.2(c)). If $R_{\max}$ were fixed at 1, according to Sec.3.2, the expected compression ratio $R$ would always remain below 1, preventing the model from switching into the full-rank regime.

### 3.4 OBJECTIVE FUNCTION

By jointly considering model performance, compression ratio, and guidance loss, the objective function of ARA is defined as:

$$\min_{\{\boldsymbol{\alpha}_i\}_{i=1}^N} \mathcal{L} = \underbrace{\operatorname{CE}\big(f(\boldsymbol{x}, \{\boldsymbol{\alpha}_i\}_{i=1}^N), \boldsymbol{y}\big)}_{\mathcal{L}_m} + \lambda_1 \underbrace{\frac{1}{N} \sum_{i=1}^N L_{g,i}}_{\mathcal{L}_g} + \lambda_2 \underbrace{\left( \frac{1}{C_t} \sum_{i=1}^N C(\boldsymbol{\alpha}_i) - R_{\text{target}} \right)^2}_{\mathcal{L}_c} \tag{9}$$

Here, $\{\boldsymbol{\alpha}_i\}_{i=1}^N$ denotes the set of trainable parameters for all $N$ compressible modules, $f$ is the LLM function, and $(\boldsymbol{x}, \boldsymbol{y})$ are the input text and its corresponding label. CE is the standard cross-entropy loss between the model prediction and ground-truth label. $C_t$ is the total parameter count of the original model, $C(\boldsymbol{\alpha}_i)$ represents the parameter size of the $i$-th module determined by its parameters $\boldsymbol{\alpha}_i$, and $R_{\text{target}}$ is the target compression ratio. The terms $\lambda_1$ and $\lambda_2$ are hyperparameters.

In Equation 9, $\mathcal{L}_m$ preserves the model's task performance, $\mathcal{L}_g$ guides critical modules to switch to their original full-rank weights, and $\mathcal{L}_c$ penalizes the deviation of the achieved compression ratio from the target via an $\ell_2$ loss. Since this soft constraint cannot guarantee an exact match, ARA rescales the compression ratios of all modules proportionally after training to precisely meet the target.

## 4 EXPERIMENTS

### 4.1 EXPERIMENT SETTINGS

**Baselines.** For a systematic comparison with various types of methods, we compare ARA with two groups of methods: (1) general compression ratio allocation methods based on model characteristics, including DLP (Chen et al., 2025) and FARMS (Hu et al., 2025), where the compression ratio is allocated at the transformer-layer level; (2) rank allocation methods specifically designed for SVD compression, including the heuristic method STRS (Yuan et al., 2024a), the mask training method ARS (Gao et al., 2024), and Dobi-SVD$_1$ (Qinsi et al., 2025). Since Dobi-SVD involves the complete SVD compression pipeline, we adopt Dobi-SVD$_1$ to represent its rank allocation algorithm.

**Models and Datasets.** We conduct experiments on models from both the LLaMA (Touvron et al., 2023) and Qwen (Yang et al., 2025) families. Following prior work that extensively evaluated LLaMA2 models, we select LLaMA2-7B and LLaMA2-13B. To further validate the generality of our method, we additionally test on Qwen3-8B and Qwen3-14B. We evaluate language modeling ability on WikiText-2 (Merity et al., 2017) and C4 (Raffel et al., 2020), and assess zero-shot task performance on ARC-Easy, ARC-Challenge (Clark et al., 2018), HellaSwag (Zellers et al., 2019), OpenBookQA (Mihaylov et al., 2018), WinoGrande (Sakaguchi et al., 2021), MathQA (Amini et al., 2019), and PIQA (Bisk et al., 2020).

**Experiment Setup.** Consistent with prior work, we compress the linear modules within transformer layers, where the compression ratio is defined as the ratio of the compressed model's parameters to

Table 1: Perplexity and zero-shot task results of LLaMA2-7B and Qwen3-8B models under 80% and 60% compression ratios. Bold indicates the optimal results.

| Model | Comp. Rate | Method | Perplexity ↓ | | Zero-shot Task Accuracy (%) ↑ | | | | | | | Avg. (%) ↑ |
|---|---|---|---|---|---|---|---|---|---|---|---|---|
| | | | Wiki2 | C4 | ARC-e | ARC-c | Hella | OBQA | Wino | MathQA | PIQA | |
| LLaMA2 -7B | - | Dense | 5.47 | 7.26 | 76.30 | 43.34 | 57.14 | 31.40 | 69.14 | 28.17 | 78.07 | 54.79 |
| | 80% | Uniform | 8.38 | 20.13 | 55.93 | 25.77 | 39.47 | 25.40 | 60.30 | 23.72 | 66.16 | 42.39 |
| | | DLP | 8.45 | 18.76 | 59.68 | 27.65 | 40.70 | 26.80 | 61.33 | 23.18 | 67.19 | 43.79 |
| | | FARMS | 8.38 | 19.74 | 55.26 | 25.34 | 39.66 | 26.80 | 61.80 | 23.52 | 65.56 | 42.56 |
| | | STRS | 7.80 | 17.41 | 53.91 | 26.02 | 41.02 | 26.00 | 60.54 | 24.02 | 66.21 | 42.53 |
| | | ARS | 8.07 | 18.46 | 62.79 | 30.12 | 42.21 | 27.00 | 65.59 | 24.19 | 69.21 | 45.87 |
| | | Dobi-SVD$_1$ | 7.92 | 16.26 | 58.92 | 25.94 | 39.93 | 27.00 | 60.93 | 23.89 | 66.27 | 44.40 |
| | | ARA | **6.42** | **10.10** | **74.28** | **38.82** | **50.89** | **30.60** | **67.48** | **27.54** | **75.14** | **52.11** |
| | 60% | Uniform | 16.14 | 71.52 | 39.48 | 21.33 | 30.70 | 19.00 | 55.80 | 21.98 | 58.32 | 35.23 |
| | | DLP | 14.24 | 55.85 | 38.68 | 22.10 | 32.13 | 19.80 | 57.06 | 22.04 | 59.63 | 35.92 |
| | | FARMS | 15.49 | 65.58 | 39.44 | 22.18 | 31.05 | 19.40 | 55.56 | 21.91 | 57.89 | 35.35 |
| | | STRS | 18.12 | 82.58 | 37.04 | 21.76 | 29.10 | 15.60 | 53.51 | 21.68 | 56.96 | 33.66 |
| | | ARS | 14.49 | 59.65 | 40.57 | 40.57 | 31.75 | 19.80 | 57.46 | 22.35 | 59.14 | 36.21 |
| | | Dobi-SVD$_1$ | 12.95 | 41.93 | 46.04 | 22.78 | 32.85 | 20.20 | 56.91 | 22.08 | 61.97 | 37.55 |
| | | ARA | **10.85** | **27.57** | **56.06** | **26.19** | **35.91** | **23.40** | **61.01** | **22.98** | **63.93** | **41.35** |
| Qwen3 -8B | - | Dense | 9.72 | 15.42 | 83.54 | 55.97 | 57.13 | 31.00 | 67.80 | 49.61 | 76.88 | 60.28 |
| | 80% | Uniform | 13.99 | 37.45 | 68.73 | 40.87 | 44.34 | 28.40 | 66.30 | 30.02 | 69.91 | 49.80 |
| | | DLP | 15.67 | 42.60 | 68.14 | 42.32 | 43.60 | 28.00 | 63.85 | 31.56 | 68.28 | 49.39 |
| | | FARMS | 14.44 | 38.72 | 69.53 | 41.47 | 44.27 | 27.80 | 66.06 | 30.89 | 69.64 | 49.95 |
| | | STRS | 143.28 | 488.53 | 36.83 | 19.37 | 29.17 | 14.20 | 52.80 | 21.47 | 59.14 | 33.28 |
| | | ARS | 15.92 | 50.30 | 52.65 | 28.41 | 37.48 | 22.40 | 57.38 | 23.52 | 65.61 | 41.07 |
| | | Dobi-SVD$_1$ | 12.45 | 30.81 | 72.39 | 43.52 | 45.62 | 29.60 | 65.19 | 32.23 | 71.33 | 51.41 |
| | | ARA | **10.75** | **20.05** | **80.43** | **52.39** | **49.34** | **29.80** | **67.01** | **37.96** | **73.78** | **55.81** |
| | 60% | Uniform | 29.45 | 132.00 | 40.45 | 21.16 | 31.45 | 15.60 | 54.38 | 22.51 | 59.74 | 35.04 |
| | | DLP | 27.78 | 117.86 | 42.72 | 23.21 | 32.16 | 17.80 | 56.75 | 23.25 | 60.39 | 36.61 |
| | | FARMS | 34.50 | 168.17 | 40.78 | 21.50 | 31.00 | 16.00 | 53.59 | 22.68 | 58.98 | 34.93 |
| | | STRS | NaN | NaN | 27.15 | 18.69 | 26.68 | 11.60 | 49.41 | 21.04 | 51.96 | 29.50 |
| | | ARS | 52.00 | 235.32 | 31.78 | 18.94 | 28.75 | 13.20 | 50.99 | 21.04 | 56.69 | 31.63 |
| | | Dobi-SVD$_1$ | 18.65 | 64.08 | 55.13 | 29.52 | 35.90 | 24.40 | 59.59 | 23.92 | 63.60 | 41.72 |
| | | ARA | **16.34** | **47.71** | **59.01** | **30.97** | **36.24** | **22.40** | **57.46** | **25.03** | **65.72** | **42.40** |

the original parameter count. Without loss of generality, we use SVD-LLM (Wang et al., 2025) as the uniform method. For training, we use the first shard of the C4 dataset and randomly select 256 samples, each containing 512 tokens. The hyperparameters $\lambda_1$, $\lambda_2$, and $D$ are all set to 100. We adopt AdamW (Loshchilov & Hutter, 2019) optimizer with a learning rate of $1 \times 10^{-3}$ and train for 10 epochs. All experiments are implemented in PyTorch, and zero-shot evaluations are conducted using the LM-Evaluation-Harness framework (Sutawika et al., 2023).

## 4.2 EXPERIMENTAL RESULTS

**Performance Comparison** We evaluate ARA at two compression ratios, 80% and 60%. The results on LLaMA2-7B and Qwen3-8B are shown in Table 1. Compared with state-of-the-art methods, ARA consistently achieves substantial improvements in both perplexity and zero-shot performance. At 80% compression, ARA reduces perplexity on WikiText-2 and C4 by 1.96 and 10.03, respectively, over the uniform compression baseline, while the best existing method achieves only marginal improvements. On zero-shot tasks, ARA yields a gain of 9.72 percentage points, significantly surpassing the strongest prior method. At 60% compression, ARA maintains its superiority, further reducing perplexity on both WikiText-2 and C4 while improving zero-shot accuracy. These trends hold on the Qwen3 model, where ARA again delivers substantial performance gains, validating the effectiveness of our adaptive rank allocation strategy.

To further examine its scalability, we extend our evaluation to larger models, including LLaMA2-13B and Qwen3-14B. As shown in Table 2, ARA achieves the best performance across both per-

Table 2: Performance of LLaMA2-13B and Qwen3-14B models under 80% compression ratios.

| Model | Method | Perplexity ↓ | | Zero-shot Task Accuracy (%) ↑ | | | | | | | Avg. (%) ↑ |
|---|---|---|---|---|---|---|---|---|---|---|---|
| | | Wiki2 | C4 | ARC-e | ARC-c | Hella | OBQA | Wino | MathQA | PIQA | |
| LLaMA2 -13B | Dense | 4.88 | 6.73 | 79.42 | 48.29 | 60.07 | 35.20 | 72.22 | 32.19 | 79.11 | 58.07 |
| | Uniform | 6.66 | 14.99 | 67.76 | 32.85 | 44.29 | 29.00 | 67.32 | 25.73 | 71.16 | 48.30 |
| | DLP | 6.73 | 14.08 | 67.51 | 33.28 | 45.27 | 29.20 | 67.40 | 26.13 | 71.71 | 48.64 |
| | FARMS | 6.64 | 14.63 | 67.47 | 32.51 | 44.40 | 28.60 | 67.40 | 25.56 | 70.89 | 48.12 |
| | STRS | 15.89 | 63.09 | 41.37 | 19.28 | 29.11 | 15.40 | 52.01 | 21.14 | 58.00 | 33.76 |
| | Dobi-SVD$_1$ | 6.37 | 12.84 | 68.56 | 33.96 | 46.01 | 30.20 | 68.11 | 25.80 | 73.01 | 49.38 |
| | ARA | **5.55** | **9.13** | **77.61** | **43.00** | **53.12** | **32.60** | **70.48** | **28.11** | **77.04** | **54.57** |
| Qwen3 -14B | Dense | 8.64 | 13.81 | 84.13 | 58.62 | 60.96 | 34.80 | 73.01 | 56.45 | 80.09 | 64.01 |
| | Uniform | 12.33 | 37.02 | 74.12 | 48.04 | 48.16 | 30.60 | 69.53 | 37.02 | 72.14 | 54.23 |
| | DLP | 14.50 | 46.16 | 73.44 | 45.56 | 47.11 | 33.00 | 69.30 | 35.85 | 71.27 | 53.65 |
| | FARMS | 12.54 | 37.75 | 73.19 | 46.84 | 47.92 | 30.80 | 69.46 | 36.62 | 72.31 | 53.88 |
| | STRS | 56.55 | 321.64 | 38.93 | 21.59 | 30.72 | 14.60 | 52.80 | 21.94 | 59.03 | 34.23 |
| | Dobi-SVD$_1$ | 11.38 | 28.83 | 71.84 | 43.09 | 48.52 | 31.80 | 67.01 | 35.81 | 72.14 | 52.89 |
| | ARA | **9.30** | **17.90** | **80.09** | **53.58** | **52.79** | **32.80** | **72.85** | **43.85** | **76.44** | **58.92** |

plexity and zero-shot tasks, confirming its strong generalization capability across model sizes. It is worth emphasizing that ARA exhibits superior robustness over existing methods. The general-purpose approaches, DLP and FARMS, suffer from unsatisfactory perplexity at a low compression ratio of 80%. Moreover, STRS and ARS demonstrate substantially inferior performance compared to Uniform on the Qwen3 series, suggesting limited scalability across model families. Although Dobi-SVD$_1$ stands as the strongest baseline, its zero-shot accuracy on Qwen3-14B still falls short of Uniform. In contrast, ARA is the only method that consistently achieves stable performance gains across all experiments, thereby underscoring its remarkable robustness.

**Combination with Quantization** Quantization is an important approach for reducing model size. To verify the compatibility of our method with quantization, we apply GPTQ (Frantar et al., 2022) to quantize the 80% compressed LLaMA2-7B model to 4-bit. The results are shown in Table 3. Prior to quantization, the SVD model is fine-tuned to recover performance using the settings described in Sec.4.3. Compared with pure quantization methods of the same memory budget, the combination of ARA and quantization achieves lower perplexity and higher zero-shot accuracy. This result indicates that integrating ARA with quantization can effectively mitigate the performance degradation caused by extreme compression, thus demonstrating its potential for creating highly compressed yet performant models.

Table 3: Comparison of SVD-Quantization Hybrid and Pure Quantization for LLaMA2-7B under a 3GB Memory Budget

| Method | Perplexity↓ | | Avg. (%) ↑ |
|---|---|---|---|
| | Wiki2 | C4 | |
| Uniform(4-bit) | 11.99 | 16.55 | 43.11 |
| LLaMA2-7B(3-bit) | 8.36 | 10.75 | 48.30 |
| ARA(4-bit) | **7.54** | **10.73** | **49.53** |

Table 4: Comparison against prior structural compression methods on LLaMA2-7B at 80% compression ratio. The performance of the pruning methods are derived from Tian et al. (2025).

| Model | Method | PPL ↓ | Zero-shot Task Accuracy (%) ↑ | | | | | | Avg. (%) ↑ |
|---|---|---|---|---|---|---|---|---|---|
| | | Wiki2 | PIQA | WinoG. | HellaS. | ARC-e | ARC-c | OBQA | |
| LLaMA2 -7B | dense | 5.11 | 78.07 | 69.14 | 75.92 | 76.30 | 43.34 | 44.20 | 64.88 |
| | LLM-Pruner | 10.55 | **75.95** | 63.38 | 67.83 | 64.31 | **39.93** | 39.60 | 58.50 |
| | FLAP | 6.76 | 74.54 | 62.98 | 64.74 | 61.28 | 36.43 | 39.60 | 56.60 |
| | SliceGPT | 8.24 | 64.80 | 62.98 | 49.18 | 55.68 | 31.40 | 33.00 | 49.51 |
| | ARA | **5.99** | 75.14 | **67.48** | **68.49** | **74.28** | 38.82 | **42.00** | **61.04** |

**Comparison with Structured Pruning.** In addition to being orthogonal to quantization, SVD is also orthogonal to pruning methods. Here, we compare ARA with mainstream structured pruning approaches, namely LLM-Pruner (Ma et al., 2023), FLAP (An et al., 2024), and SliceGPT (Ashkboos et al., 2024), with the results summarized in Table 4. As shown in the table, under a 80% compression ratio on the LLaMA2-7B model, ARA reduces perplexity by 11.4 relative percentage points compared to the state-of-the-art structured pruning methods, while also improving zero-shot task accuracy by 2.54 absolute percentage points.

Table 5: Performance comparison of different mask generation methods for LLaMA2-7B with the same training objective.

| Comp. Rate | Method | epochs | perplexity↓ | | Avg.(%) ↑ |
|---|---|---|---|---|---|
| | | | wiki2 | c4 | |
| 80% | ARS | 1953 | 8.22 | 19.23 | 43.22 |
| | Dobi-SVD$_1$ | 20 | 7.92 | 16.26 | 44.40 |
| | ARA | 10 | **7.83** | **15.76** | **44.92** |
| 60% | ARS | 2932 | 13.93 | 56.43 | 36.08 |
| | Dobi-SVD$_1$ | 20 | 12.95 | 41.93 | 37.55 |
| | ARA | 10 | **12.57** | **39.71** | **38.59** |

Table 6: LoRA fine-tuning results of LLaMA2-7B model under different compression ratios.

| Comp. Rate | Method | perplexity↓ | | Avg.(%) ↑ |
|---|---|---|---|---|
| | | wiki2 | c4 | |
| - | dense | 5.47 | 7.26 | 54.79 |
| 80% | ARA | 6.42 | 9.99 | 52.11 |
| | w. LoRA | **6.37** | **9.31** | **54.23** |
| 60% | ARA | 10.85 | 27.57 | 41.35 |
| | w. LoRA | **9.98** | **14.74** | **46.80** |

## 4.3 ABLATION STUDY

**Effectiveness of Mask Generation** To evaluate the effectiveness of our mask generation strategy, we conduct experiments using only the $L_m$ and $L_c$ terms in Eq. (9), while removing the $L_g$ term. In this setting, our loss function is equivalent to that of ARS (Gumbel-Sigmoid mask) and Dobi-SVD$_1$ (tanh-based mask), which provides a fair comparison across methods. We uniformly train models decomposed via SVD with unchanged parameter counts, thereby eliminating the effect of the mask scope. Each training epoch consists of 256 samples, and other hyperparameters are kept at their default values. The results are shown in Table 5. Compared with tanh-based and Gumbel-Sigmoid masks, our proposed mask generation method achieves better performance within fewer training epochs. Here, the ARS performs the worst, highlighting the importance of maintaining the monotonicity of the mask. Meanwhile, the results of Dobi-SVD$_1$ also indicate that parameter updates should be influenced by the global singular values.

**Fine-tuning After Compression** Compared with previous approaches, ARA already demonstrates strong performance. To further explore their potential, we fine-tune all compressed modules with LoRA (Hu et al., 2022) using the Alpaca dataset (Taori et al., 2023), and after fine-tuning, we merge the LoRA parameters back into the corresponding matrices. Table 6 reports the results on the LLaMA2-7B model. At a compression ratio of 80%, fine-tuning yields a significant improvement in zero-shot accuracy, exhibiting only a minor performance gap relative to the original model. These findings indicate that ARA effectively preserves critical components of the model, thereby facilitating subsequent performance recovery through fine-tuning.

## 5 CONCLUSION

In this work, we proposed the Adaptive Rank Allocation (ARA) method to address the rank allocation problem in SVD-based compression of large language models. ARA leverages a specially designed mask to better capture the importance of singular values, while introducing additional loss terms to mitigate the non-smoothness of the loss landscape when switching between low-rank and full-rank representations. Extensive experiments on the LLaMA2 and Qwen3 model families demonstrate that ARA consistently achieves superior performance in both perplexity and zero-shot tasks, significantly surpassing state-of-the-art baselines. These results provide a new perspective on rank allocation in SVD-based model compression and highlight the potential of ARA as a practical and effective solution.

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

# A APPENDIX

## A.1 PSEUDOCODE

---

**Algorithm 1** Overall algorithm of ARA

---

1: **Input:** calibration dataset $\mathcal{X}$, pre-trained LLM model $\{W^l\}_{l=1}^N$, target compression ratio $R_{\text{target}}$.
2: **Output:** compressed model.
3:                                                        $\triangleright$ 1. Pre-computation and Initialization
4: **for** $l$ in $\{1, 2, \ldots, N\}$ **do**
5:     Calculate SVD components $U^l, \Sigma^l, V^l, S^l$ for weight matrix $W^l$ using input from $\mathcal{X}$.
6:     Initialize trainable parameters $\boldsymbol{\alpha}^l$ on the probability simplex.
7: **end for**
8:                                                                 $\triangleright$ 2. Training Loop
9: **for** each training step **do**
10:     Initialize total guidance loss $\mathcal{L}_g \leftarrow 0$, current total parameters $C_{current} \leftarrow 0$.
11:     **for** $l$ in $\{1, 2, \ldots, N\}$ **do**
12:         Calculate probability mask $\boldsymbol{p}^l$ from $\boldsymbol{\alpha}^l$ by equation 2.
13:         Calculate compression ratio $R^l$ from $\boldsymbol{p}^l$ by equation 3.
14:         Determine binary mask $\boldsymbol{m}^l$ from $R^l$ by equation 4.
15:         Calculate guidance metric $G_{R^l}$ and loss $L_{g,l}$ by equation 6 and equation 7.
16:         Determine effective weight matrix $W'^l$ based on $R^l$ by equation 8.
17:         $\mathcal{L}_g \leftarrow \mathcal{L}_g + L_{g,l}$.
18:         $C_{current} \leftarrow C_{current} + \text{parameter\_count}(W'^l)$.
19:     **end for**
20:     Perform forward pass with effective weights $\{W'^l\}$ to get model loss $\mathcal{L}_m = \text{CE}(f(\mathcal{X}), \mathcal{Y})$.
21:     Calculate compression loss $\mathcal{L}_c = (C_{current}/C_{total} - R_{\text{target}})^2$.
22:     Calculate total objective $\mathcal{L} = \mathcal{L}_m + \lambda_1 \mathcal{L}_g + \lambda_2 \mathcal{L}_c$ by equation 9.
23:     Update all $\{\boldsymbol{\alpha}^l\}_{l=1}^N$ by back-propagation using equation 5 for gradients through $\boldsymbol{m}^l$.
24: **end for**
25:                                $\triangleright$ 3. Post-processing and Final Model Construction
26: Rescale all final ratios $\{R^l\}$ proportionally to meet $R_{\text{target}}$ exactly.
27: Generate final binary masks $\{M^l\}$ from rescaled ratios.
28: Construct final compressed model $\{\boldsymbol{W}'^l\}_{l=1}^N$
29: **return** compressed model

---

## A.2 RANK DISTRIBUTION.

In this section, we provide a detailed analysis of the rank distribution in LLMs. At a compression ratio of 80%, the final training results for the LLaMA2-7B and Qwen3-8B models are shown in Fig. 4(a).

The rank distribution exhibits clear differences across models. For LLaMA2-7B, the first three layers and the last two layers tend to retain relatively lower ranks. In contrast, Qwen3-8B allocates a larger rank to its first layer compared to its second and third layers, while its last two layers do not show a significant decrease. The behaviors of the middle layers also differ substantially: Qwen3-8B tends to compress more parameters in these regions. Moreover, the importance of different modules varies considerably between models. For instance, the key module plays a more critical role in Qwen3-8B, whereas up module is relatively less important.

When examining modules individually, we observe significant variation in the proportion of preserved rank. In LLaMA2-7B, the query and key projections exhibit much higher compression ratios, indicating that only a small number of singular values are sufficient to maintain their functionality. Conversely, the value, gate, and down projections are often left uncompressed, with the original matrices retained to preserve essential information, highlighting their critical role in maintaining overall model performance. Qwen3-8B demonstrates a similar trend: the value, up, and down projections appear to be the most important components.

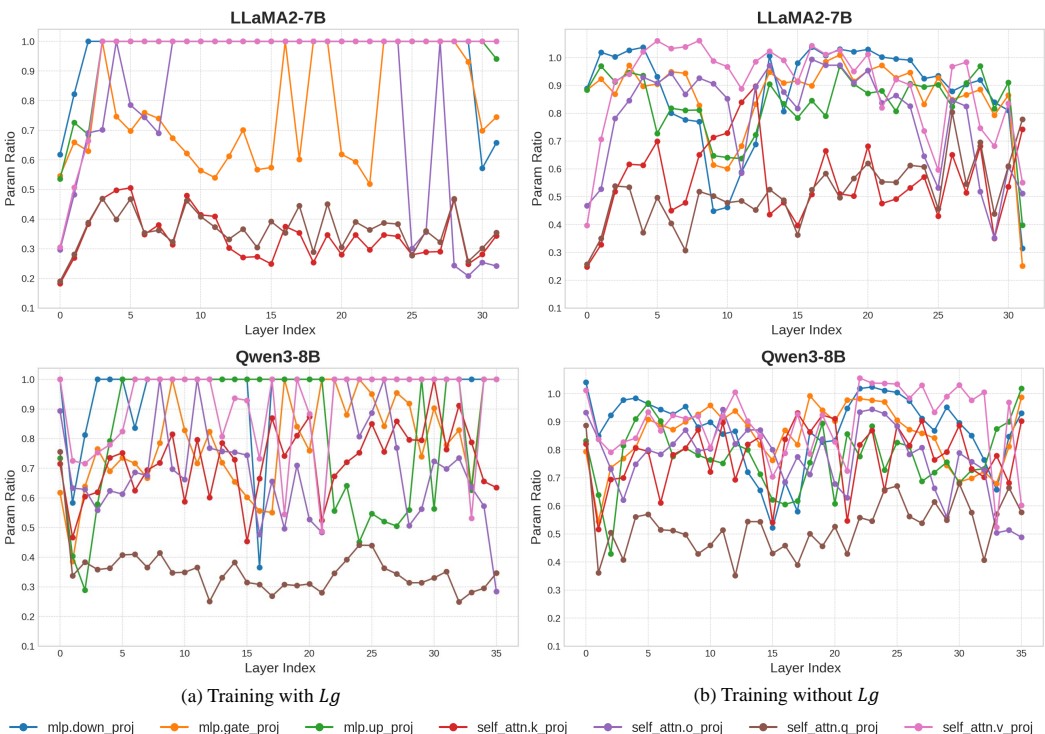

Figure 4: Parameter ratio of linear layers in LLaMA2-7B and Qwen3-8B under different training conditions.

These results further demonstrate the significant heterogeneity among different modules within LLMs, highlighting the importance of research on rank allocation.

## A.3 EFFECT OF $L_g$

The effects of the mask generation strategy are presented in Sec.4.3, and here we further investigate the role of the $L_g$ term in the loss function of Eqn.9. Fig.4(b) presents the training results when $L_g$ is excluded. As shown, without $L_g$, only a small fraction of modules reach a compression ratio greater than 1. Compared to Fig.4(a), most ranks converge to ratios smaller than 1, confirming that without switching computation modes when the compression ratio equals 1, the model tends to converge only to a suboptimal local solution. By contrast, with the guidance of $L_g$, many modules eventually choose to retain their original matrices, thereby better preserving the overall performance of the model.

## A.4 INFERENCE SPEEDUP ON HARDWARE

From a theoretical perspective, the computational compression ratio of SVD should match its parameter compression ratio. To verify this in practice, following the experimental setup of SVD-LLM (Wang et al., 2025), we evaluate the inference throughput of LLaMA2-7B on a single NVIDIA 3090 GPU under varying batch sizes and sequence lengths, measuring the number of tokens generated per second at different compression ratios. The results are shown in Fig.5.

Overall, ARA with a 60% compression ratio achieves a 1.23× throughput improvement over ARA with an 80% compression ratio when the sequence length is fixed at 32 across different batch sizes, and a 1.26× improvement when the batch size is fixed at 64 across different sequence lengths. Moreover, for the same target compression ratio, ARA consistently outperforms uniform compression: at 80% compression, ARA achieves a 1.16× higher throughput, and at 60% compression, the factor rises to 1.20×.

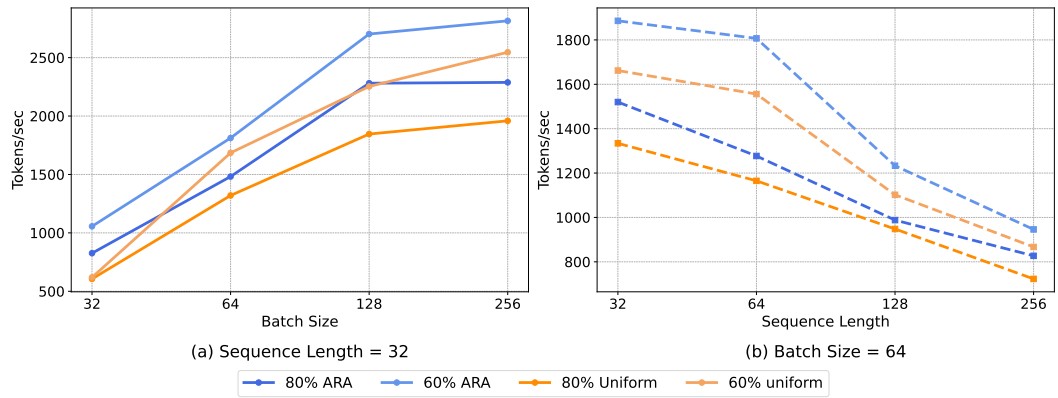

Figure 5: Throughput (tokens/sec) for LLaMA2-7B with ARA and uniform at different compression rates, under (a) varying batch sizes and (b) varying sequence lengths.

## A.5 ABLATION ON HYPERPARAMETERS

In this section, we investigate the impact of training hyperparameters on model performance. Unless otherwise specified, all experiments are conducted on the LLaMA2-7B model under an 80% compression ratio, and results are evaluated in terms of perplexity on the WikiText2 and C4 datasets.

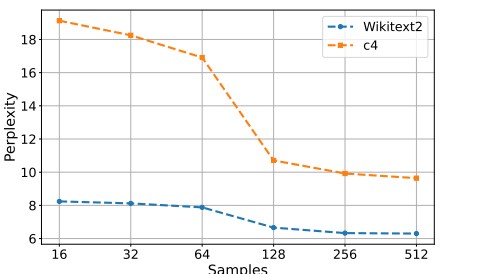

Figure 6: PPL vs. samples(epochs=10)

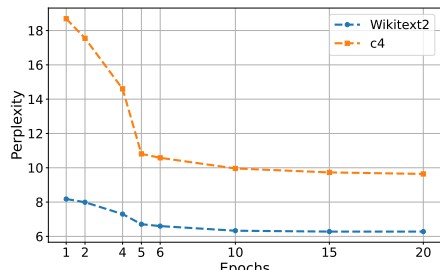

Figure 7: PPL vs. epochs(samples=256)

**Training Samples.** We first analyze the influence of the number of training samples. Fig.6 reports the perplexity after 10 training epochs with different sample sizes. When the number of samples is fewer than 128, the model perplexity decreases rapidly as the sample size increases. However, beyond 128 samples, the improvements become much less pronounced. For instance, when the sample size increases from 128 to 256, the perplexity is reduced by 0.33 on WikiText2 and 0.79 on C4; in contrast, further increasing the sample size from 256 to 512 yields only marginal gains of 0.03 and 0.28, respectively. These results suggest that the model is sufficiently trained with 256 samples, and additional data provides little benefit. Hence, we adopt 256 samples as the default setting.

**Training Epochs.** We next examine the effect of training epochs. Fig.7 presents the perplexity under different numbers of epochs. A clear performance improvement is observed at the 5th epoch, which we attribute to some modules being trained to a compression ratio of 1, thereby switching to the original dense matrix and recovering all truncated ranks in a single step. This transition leads to a notable perplexity reduction. However, beyond 10 epochs, the performance gains diminish significantly. Increasing the training duration from 10 to 20 epochs reduces perplexity by only 0.05 on WikiText2 and 0.32 on C4, which is negligible compared to earlier improvements. Therefore, we adopt 10 epochs as the default training schedule.

**Training Parameters.** As introduced in Sec.3.2, each mask in ARA is parameterized by $D$ trainable parameters and controlled through a staircase-shaped binary matrix $M \in \{0, 1\}^{D \times r}$. Specifically, we assign every $\lfloor r/D \rfloor$ columns to one step, ensuring that each step corresponds to the same number of singular values. The format of $M$ is shown as follows:

$$M = \begin{bmatrix} 1 & 1 & 0 & 0 & 0 & 0 & 0 & 0 \\ 1 & 1 & 1 & 1 & 0 & 0 & 0 & 0 \\ 1 & 1 & 1 & 1 & 1 & 1 & 0 & 0 \\ 1 & 1 & 1 & 1 & 1 & 1 & 1 & 1 \end{bmatrix}$$

The choice of $D$ determines the granularity of the mask: when $D = 1$, all singular values share the same retention probability; when $D = r$, each singular value has its own probability. We experiment with three configurations, $D \in \{10, 100, 1000\}$, and the results are reported in Tab.7. The results show that moving from $D = 10$ to $D = 100$ yields noticeable improvements in model performance, while $D = 1000$ achieves the same results as $D = 100$. This indicates that $D = 100$ provides sufficient granularity to control the mask, and further increasing the number of parameters does not lead to additional gains. Therefore, we adopt $D = 100$ as the default setting.

$\lambda_1$ **and** $\lambda_2$ In equation 9, the hyperparameters $\lambda_1$ and $\lambda_2$ control the weights of the guidance loss and the compression rate loss, respectively. We investigate the impact of these parameters on model performance. To simplify the analysis, we set $\lambda_1 = \lambda_2 = \lambda$ and explore various values. As shown in Table 8, the results indicate that the final performance is not highly sensitive to the choice of $\lambda$. Therefore, we select the best accuracy setting: $\lambda_1 = \lambda_2 = 100$ as the default setting.

<table>
<tr><td colspan="4">Table 7: Ablation experiment for D</td></tr>
<tr><td>D</td><td>10</td><td>100</td><td>1000</td></tr>
<tr><td>Wiki2</td><td>6.53</td><td>6.42</td><td>6.42</td></tr>
<tr><td>C4</td><td>10.32</td><td>10.10</td><td>10.10</td></tr>
</table>

Table 7: Ablation experiment for D

| D | 10 | 100 | 1000 |
|---|-----|------|------|
| Wiki2 | 6.53 | 6.42 | 6.42 |
| C4 | 10.32 | 10.10 | 10.10 |

Table 8: Ablation experiment for $\lambda$

| $\lambda$ | 50 | 100 | 200 |
|---|-----|------|------|
| Wiki2 | 6.33 | 6.42 | 6.47 |
| C4 | 9.98 | 10.10 | 10.20 |
| Avg. Acc. (%) | 51.95 | 52.11 | 51.87 |

## A.6 BASELINE HYPERPARAMETER SETTINGS

Since the baselines we adopt do not provide native implementations under our framework, we list here the key hyperparameters used in our experiments to control the final compression ratio range across modules. For fairness and reproducibility, we mainly follow the default settings reported in the original papers or select one from the available options.

- **DLP**: $\alpha = 0.15$
- **FARMS**: $\epsilon = 0.3$
- **STRS**: set of potential truncation ratios $\{0.1, 0.2, \ldots, 0.9\}$
- **ARS**: we use the third-party implementation[1] with the following settings: `layer_type = simple, r_loss = simple, lambda_scale = 1.0, gamma_scale = 0., beta_scale = 0.5`
- **Dobi-SVD**[1]: learning rate = 5, $\alpha = 200$, epochs = 20, nsamples = 256

## A.7 USE OF LLMS.

Large language models (LLMs) played an important role throughout this work. During the research stage, LLMs assisted in clarifying fundamental concepts. In the experimentation stage, they were primarily used to help implement code for specific functionalities. Finally, during the writing stage, LLMs contributed to content refinement and supported tasks such as generating LaTeX tables.

---

[1]https://github.com/sidhantls/adaptive-rank-selection-svd

