# OpenReview forum: "ARA: Adaptive Rank Allocation for Efficient Large Language Model SVD Compression"
_ICLR.cc/2026/Conference — ICLR 2026 Conference Withdrawn Submission_

### Official Review · Reviewer_pWMm · 2025-10-30

**Soundness:** 3
**Presentation:** 3
**Contribution:** 2
**Rating:** 4
**Confidence:** 4

**Summary:**

This paper proposes Adaptive Rank Allocation (ARA), a method for allocating different compression ranks across linear modules in LLM compression via SVD. The key contributions are: (1) a staircase mask design that enables efficient rank allocation with monotonicity and global update influence, (2) a guidance loss function that handles the discontinuity at compression ratio = 1 by allowing modules to retain full-rank matrices when SVD would be parameter-inefficient, and (3) strong empirical results showing significant improvements over baselines. I have some concerns about the experiment and relationship between this paper and previous work.

**Strengths:**

1. Strong Empirical Result.
2. Novel Treatment of R≥1 Case: The guidance loss Lg and dynamic computational flow (Equation 8) address a genuine problem that prior mask-based methods overlook. This is a valuable contribution.
3. Improved Mask Design: The staircase binary matrix parameterization ensures monotonicity while avoiding vanishing gradients (unlike tanh-based masks) and maintaining global receptive field (unlike Gumbel-Sigmoid masks). The technical design is sound.

**Weaknesses:**

1. Missing Baselines and Citations.
The paper should cite the following two papers in both literature review and experimental comparisons:
1) TFWSVD—EMNLP 2022, titled "Numerical Optimizations for Weighted Low-rank Estimation on Language Model
2)  RankDyna— EMNLP 2023 findings, Dynamic Low-rank Estimation for Transformer-based Language Models

TFWSVD is the follow-up of FWSVD, in a more accurate way. More importantly, that paper proposes a Fisher information variance metric φ(W) that predicts when SVD will fail for a given matrix, which is conceptually very similar to what ARA's guidance loss is trying to achieve. Matrices with high φ(W) should be the ones where ARA's guidance loss activates and chooses full rank.
Second, RankDyna performs dynamic rank selection for SVD compression of LLMs. The key difference is that RankDyna learns ranks during fine-tuning while ARA learns them separately on calibration data. The computation cost may be close, please consider add it as a baseline.

2.   Technical concerns
The straight-through estimator used to backpropagate through the binary mask introduces gradient bias that isn't discussed. The paper doesn't analyze training stability, convergence properties, or sensitivity to initialization.
3. Experiment concerns.
The paper mentions: "ARA rescales the compression ratios of all modules proportionally after training to precisely meet the target." Should be discussed more.

**Questions:**

1. Is the guidance loss learning the same thing that φ(W) (proposed in TFWSVD) already measures?
2. If not, can we consider some analysis based on φ(W)? For example: φ(W)-Based Threshold for Activation, φ(W)-Weighted Guidance Loss, φ(W)-Based Initialization?

---

> ### Author Response · Authors · 2025-11-16
>
> Thank you for your valuable comments and suggestions on our work. We will further improve our work based on your recommendations.
>
> **Regarding W1, Q1, and Q2:** Thank you for the additions to our work. We will add corresponding descriptions and results in the related work and experimental sections. The $\phi(w)$ in TFWSVD is an indicator used to measure whether a parameter matrix should use standard SVD or TFWSVD. In contrast, our guidance loss measures whether the parameter matrix needs to be decomposed at all. Their principles and mechanisms are different, and therefore $\phi(w)$ cannot replace the role of our guidance loss.
>
> **Regarding RankDyna:** The method proposed in that paper provides a valuable perspective on our research problem. Since the paper is not open-sourced, we compare our results with GRASP [1], its improved version for large language models. The results are as follows:
>
> **Table 1: Comparison of results at 80% compression rate.**
> | Model | Method | Wikitext2 PPL | Acc Avg. |
> | :--- | :--- | :--- | :--- |
> | LLaMA2-7B | GRASP | 10.19 | 0.44 |
> | LLaMA2-7B | **ARA (Ours)** | **6.42** | **0.52** |
> | LLaMA2-13B | GRASP | 9.59 | 0.47 |
> | LLaMA2-13B | **ARA (Ours)** | **5.55** | **0.56** |
>
> The results in the table show that ARA, even without fine-tuning, still outperforms RankDyna-type methods. RankDyna directly decomposes the original parameter matrix $W$, which often leads to significant performance degradation. In large models, the cost of recovering this lost performance is high. Therefore, our activation-aware decomposition is superior to the RankDyna method.
>
> **Regarding W2:** Thank you for your concern about the model training process. STE (Straight-Through Estimator) is a widely used method to convert non-differentiable masks into differentiable ones, achieving a good bias-variance trade-off [2]. For model stability, we trained on different datasets (C4, Wikitext2, Alpaca) and also with different random seeds on the C4 dataset. The results are as follows:
>
> **Table 2: Results on different training datasets (at 80% compression).**
> | Metric | Baseline | Trained on Wikitext2 | Trained on C4 | Trained on Alpaca |
> | :--- | :--- | :--- | :--- | :--- |
> | Wikitext2 PPL | 8.38 | 6.33 | 6.42 | 6.81 |
> | C4 PPL | 20.13 | 10.75 | 10.10 | 11.53 |
> | Acc Avg.(%) | 42.39 | 50.85 | 52.11 | 51.13 |
>
> **Table 3: Results on C4 dataset with different random seeds (at 80% compression).**
> | Seed | Wikitext2 PPL | Acc Avg.(%) |
> | :--- | :--- | :--- |
> | 3 | 6.42 | 52.11 |
> | 42 | 6.41 | 52.01 |
> | 133 | 6.45 | 52.14 |
> | 233 | 6.47 | 51.87 |
>
> As can be seen, different datasets have some impact on the results, but all show significant improvements compared to the baseline. The random seed has almost no effect on the results.
>
> Regarding model convergence, **Figure 7 in the Appendix** shows the curve for PPL versus training epochs. As training progresses, the PPL stably decreases. The improvement becomes minimal after 10 epochs, indicating that the training tends to converge. We ultimately chose 10 epochs.
>
> Regarding the initialization of $\alpha$: For the dimensionality of $\alpha$, **Table 7 in the Appendix** provides a sensitivity analysis on the dimension $D$, which shows that $D$ has minimal impact on the results. As for the value initialization of $\alpha$, we use a fixed initialization method where the initial compression rate equals the target compression rate. This process introduces no randomness and is therefore stable.
>
> **Regarding W3:** We first initialize the model at the target compression rate and then proceed with training. Since the compression ratio of each module fluctuates during training, we cannot guarantee the final compression rate will be exactly equal to the target. In practice, there is a minor discrepancy (<1%) between the final achieved compression rate and the target. To precisely meet the target, we proportionally scale the compression ratio of each compressed module. The final adjustment is minimal. Specifically, assume the target compression rate is $R_t$, the current overall compression rate is $R_c$, the total number of parameters is $P_t$, and the number of parameters in SVD-decomposed matrices is $P_{svd}$ (total parameters minus non-decomposed matrix parameters). If a specific SVD matrix has a current compression rate of $r_c$, its final rate $r'$ is adjusted as:
> $$r' = r_c \times \frac{R_t \times P_t + P_{svd} - P_t}{P_{svd} \times R_c}$$
> We will detail this process in the final version.
>
> ---
> ### References
> [1] GRASP: Replace Redundant Layers with Adaptive Singular Parameters for Efficient Model Compression (EMNLP, 2025)
>
> [2] Estimating or Propagating Gradients Through Stochastic Neurons

---

### Official Review · Reviewer_6EHe · 2025-10-31

**Soundness:** 2
**Presentation:** 2
**Contribution:** 2
**Rating:** 2
**Confidence:** 4

**Summary:**

This paper introduces Adaptive Rank Allocation (ARA), a method that learns layer-wise ranks for large-language-model SVD compression. ARA employs a trainable monotonic mask generated by a simplex-constrained vector and a staircase binary matrix; a guidance loss automatically switches layers to their full-rank weights when low-rank decomposition is parameter-inefficient. Trained on small calibration data with straight-through estimation, ARA produces a discrete binary mask that exactly meets a target compression ratio. Tests on LLaMA-2 and Qwen at 60% and 80% compression report lower perplexity and higher zero-shot accuracy than uniform truncation and several allocation baselines.

**Strengths:**

- Converts the discrete rank-selection problem into a continuous, simplex-constrained optimisation that is easy to train with standard back-propagation.
- The proposed staircase-mapping mask ensures monotonicity with respect to singular values, preserving the theoretical optimality of the Eckart–Young theorem and avoiding the instability of Gumbel-Sigmoid or tanh masks.
- The framework remains orthogonal to pruning and quantization, and can be combined with both for further efficiency gains.

**Weaknesses:**

- The monotonic mask enforces that singular values with larger magnitudes are always prioritized. As a result, ARA cannot invert or locally re-weight the importance of individual singular directions—only adjust the global cutoff boundary. This restricts its ability to capture task-critical, low-energy directions that matter more to downstream performance than to reconstruction error.
- The optimization objective remains dominated by spectral energy preservation and cross-entropy loss. It does not directly model task sensitivity or gradient-based importance, leading to suboptimal trade-offs when “energy ≠ task importance”.
- The experimental design compares different allocation strategies using only the data-aware SVD algorithm from SVD-LLM. While this ensures consistency, it fails to account for the algorithmic contributions of competing methods, potentially disadvantaging approaches that integrate allocation with novel compression techniques. A more equitable comparison would involve contrasting SVD+ARA against methods like Dobi-SVD that incorporate both algorithmic innovations and allocation strategies.
- The largest tested model (14B parameters) falls short of demonstrating scalability to larger-scale models (e.g., 70B), which are more representative of real-world deployment scenarios.

**Questions:**

- The evaluation is limited to LLaMA and Qwen architectures, which share substantial structural similarities. It remains unclear whether the method generalizes to diverse model architectures, particularly MoE models.
- It is unclear whether this method can be extended to other low-rank decomposition algorithms besides SVD.

---

> ### Author Response · Authors · 2025-11-16
>
> Thank you for your valuable comments and suggestions on our work. We will further improve the paper according to your feedback. We hope that our responses can address your concerns.
>
> ### W1
> Your observation that the retained singular values after matrix factorization may not necessarily follow the order of singular value magnitude is very insightful. In fact, some prior works, such as RankDyna [1] and GRASP [2], are specifically designed to address this problem. However, these methods decompose the original parameter matrix W, which is task-independent, and therefore the singular values that should be retained often do not align with the magnitude order of W’s own singular spectrum.
>
> Recent SVD-based methods have already taken task input into account. For example, FWSVD [3] incorporates gradients of weights with respect to the task; SVD-LLM [4] and Dobi-SVD [5] explicitly consider activations X during decomposition. Under these settings, retaining singular values in descending order has become the mainstream strategy.
>
> For example, ARS [6] allocates compression rates using a Gumbel-Sigmoid mask based on the FWSVD decomposition results. Since the mask cannot guarantee that the retained values are the largest ones, ARS adds an additional alignment loss to enforce consistency between the retained singular values and the top singular values. Their results show that removing the alignment loss leads to a 0.5%–1.5% performance drop. In contrast, our mask’s internal consistency is explicitly designed based on the principle that singular values should be retained in magnitude order. Dobi-SVD also adopts such a masking strategy.
>
> To further validate the importance of retaining top singular values, we use a Gumbel-Sigmoid mask to train on SVD-LLM results. One variant keeps the largest values directly (type 1), while the other follows mask selection without enforcing order (type 2). The results below demonstrate that preserving the largest singular values is a better strategy under current SVD frameworks:
>
> | Method | WikiText2 PPL | Acc Avg.(%) |
> |--------|--------------|---------|
> | type 1 | 8.07 | 45.87 |
> | type 2 | 8.32 | 44.62 |
>
> ### W2
> Thank you for your comments regarding the theoretical foundation. Although spectral energy does not equal task accuracy, prior work and our experimental results in W1 both indicate that higher spectral energy correlates with better task performance, which justifies using spectral energy as a proxy metric. In our paper, we use the general text dataset C4, and cross-entropy loss is commonly used in pretraining to evaluate general model capability. If one wishes to optimize a specific downstream task, the loss can be simply replaced with task-specific loss based on that dataset.
>
> ### W3
> Thank you for your suggestions on experimental settings. For baseline comparison, STRS and ARS are implemented using stronger decomposition backbones (SVD-LLM instead of ASVD or FWSVD). For Dobi-SVD, besides its allocation strategy, it includes a decomposition method parallel to SVD-LLM and a separate orthogonal quantization strategy. We therefore compare only the decomposition-allocation result of Dobi-SVD with ours:
>
> **LLaMA2-7B, 80% compression**
>
> | Method | Wiki2 PPL | Acc Avg.(%) |
> |--------|-----------|----------|
> | Uniform | 8.38 | 42.39 |
> | Dobi-SVD | 9.39 | 40.83 |
> | **ARA** | **6.42** | **52.11** |
>
> In fact, Dobi-SVD performs worse than baseline under this setting, and its improvements mostly come from orthogonal quantization rather than allocation strategy. In our work, we apply a stronger decomposition method to Dobi-SVD.
>
> ### W4
> The results of LLaMA2-70B at 80% compression are as follows
>
> | Method | Wiki2 PPL | Avg. Acc.(%) |
> |--------|-----------|----------------|
> | dense | 3.12 | 63.72 |
> | uniform | 5.96 | 53.39 |
> | **ARA** | **4.45** | **59.40** |
>
> This result is consistent with the 7B and 13B models: ARA significantly improves performance through adaptive allocation of compression rates.
>
> ### Q1 & Q2
> Our method provides a unified framework for compression-rate allocation for SVD-based LLM compression. Any model that employs SVD can directly apply our method to optimize rank allocation. If other low-rank decompositions share similar properties with SVD, our approach can also be extended to them.
>
> ---
>
> ## References
> [1] RankDyna, EMNLP Findings 2023 — Dynamic Low-rank Estimation for Transformer-based Language Models
> [2] GRASP, EMNLP 2025 — Replace Redundant Layers with Adaptive Singular Parameters for Efficient Model Compression
> [3] FWSVD, ICML 2021 — Language Model Compression with Weighted Low-rank Factorization
> [4] SVD-LLM, ICLR 2025 — Truncation-Aware Singular Value Decomposition for Large Language Model Compression
> [5] Dobi-SVD, ICLR 2025 — Differentiable SVD for LLM Compression and Some New Perspectives
> [6] ARS, NAACL 2024 — Adaptive Rank Selections for Low-Rank Approximation of Language Models

---

### Official Review · Reviewer_sevY · 2025-11-01

**Soundness:** 2
**Presentation:** 2
**Contribution:** 1
**Rating:** 2
**Confidence:** 3

**Summary:**

The paper proposes ARA, an adaptive rank allocation scheme for SVD-based compression of LLM linear layers. It learns per-layer binary masks over singular values with a monotone “staircase” parametrization, uses a straight-through estimator during training, and adds a full-rank guidance loss to decide when not to decompose a layer (i.e., revert to the dense matrix) under a global compression target.

**Strengths:**

- A simple, monotone mask construction that avoids vanishing gradients and enforces preference for larger singular values; the STE makes training match inference.

- Competitive numbers across multiple models and tasks, with tables showing consistent improvements over prior SVD rank-allocation methods.

**Weaknesses:**

- Incremental conceptual novelty. The core ideas that monotone masks over singular spectra and an auxiliary term that favors keeping full rank when low-rank is inefficient—extend well-known SVD truncation and mask-learning practices. The “guidance loss” formalizes a heuristic already implicit in prior pipelines that skip SVD when k(m+n)>mn. The theoretical treatment does not go beyond standard truncation analyses.

- Claims target “efficient LLMs,” yet there is no end-to-end latency/throughput study on real inference stacks (e.g., vLLM/SGLang). The work optimizes perplexity and zero-shot scores, but does not show serving wins or memory bandwidth reductions for prefill/decoding under realistic batching.

- The method requires input-aware SVD and mask training with extra losses; experiments train for multiple epochs on a small calibration split (first shard of C4, 256×512 tokens) with several new hyperparameters, which risks overfitting and weakens external validity. Ablations are narrow and do not test sensitivity to these choices.

- The objective uses a soft global-rate penalty and a post-hoc rescaling to hit the target ratio, which can change layer allocations after training.

**Questions:**

- Can you present end-to-end serving results (latency, tokens/sec, memory traffic) in vLLM or SGLang with continuous batching under mixed sequence lengths, comparing dense vs SVD-ARA models at the same accuracy?

- Does the post-training rescaling of layer rates change the final allocation materially? If so, why is training not run with the final rates to avoid train–test mismatch?

---

> ### Author Response · Authors · 2025-11-16
>
> Thank you for your valuable feedback and questions on our paper. We offer the following clarifications and additional information, which we hope will address your questions.
>
> **W1:** The issue of compression ratio allocation for SVD decomposition is indeed a widely studied problem, and "skipping SVD" has been widely used in past heuristic algorithms. However, the merits of different mask designs have not been extensively discussed. Furthermore, regarding the SVD skipping issue, although it is widely adopted in heuristic algorithms, these methods are often limited by search space constraints and the setup of prior knowledge, lacking generality. During the mask training process, to the best of our knowledge, no prior work has explicitly considered the impact of skipping SVD, which leads to a discontinuity in the loss function (as shown in Figure 2(a) of our paper). Our research designs a more efficient mask form and explicitly incorporates SVD skipping into the training process. Both of these contributions lead to significant improvements in the results (Table 1 and Table 2 in the paper). We believe these two points are highly meaningful.
>
> **W2, Q1:** Since low-rank decomposition alters the standard computation flow of LLMs, optimizing latency and throughput on inference engines (e.g., vLLM, SGLang) requires complex engineering, including but not limited to matrix operator optimization and parallel optimization. This is a separate research problem and falls outside the scope of this paper. Here, following the methodology in ESPACE (Dimensionality Reduction of Activations for Model Compression, NeuIPS 2024), we report the GEMM speedup and inference latency speedup (measured by Time to First Token, TTFT) within the Megatron-LM framework. The experimental setup uses a batch size=1, sequence length=4096, on a single NVIDIA A100 GPU. The results are as follows. The GEMM speedup is largely consistent with the compression ratio. The latency speedup is lower than the compression ratio due to other computations such as attention and activations in FFN, but it still demonstrates a clear improvement.
>
> | Model | Method | Total GEMM Latency | TTFT (Latency) |
> | :--- | :--- | :--- | :--- |
> | **Llama 2 7B** | Dense | 210ms | 368ms |
> | | 80% | 169ms (-19%) | 322ms (-12%) |
> | | 50% | 113ms (-46%) | 266ms (-28%) |
> | **Llama 2 13B** | Dense | 406ms | 643ms |
> | | 80% | 336ms (-17%) | 562ms (-13%) |
> | | 50% | 259ms (-36%) | 447ms (-31%) |
>
> **W3:** Thank you for your concern regarding the stability of our experiments. We present the final training results on different training datasets (C4, Wikitext2, Alpaca). As shown, different datasets have a certain impact on the results, but the model performance is significantly improved compared to uniform compression.
>
> **Table: Results of Llama 2-7B at 80% compression rate on different training datasets.**
> | Metric | Baseline | Trained on Wikitext2 | Trained on C4 | Trained on Alpaca |
> | :--- | :--- | :--- | :--- | :--- |
> | Wikitext2 PPL | 8.38 | 6.33 | 6.42 | 6.81 |
> | C4 PPL | 20.13 | 10.75 | 10.10 | 11.53 |
> | Acc Avg.(%) | 42.39 | 50.85 | 52.11 | 51.13 |
>
> As for the hyperparameter ablation study, we provide a detailed analysis in Appendix A.6. This includes the number of training samples, the number of epochs, the number of trainable parameters per matrix (D), and the loss coefficients $\lambda_1$ and $\lambda_2$. The experimental results indicate that our method is not highly sensitive to these hyperparameter settings.
>
> **W4, Q2:** We first initialize the model at the target compression ratio and then proceed with training. Since the compression ratio of each module fluctuates during the training process, we cannot guarantee that the final compression ratio will be exactly equal to the target. In practice, there is a minor discrepancy (<1%) between the final achieved compression ratio and the target. To precisely meet the target, we proportionally scale the compression ratio of each compressed module based on the ratio between the achieved and target values. This final adjustment is minimal and, being proportional, does not substantially alter the training results.

---

### Official Review · Reviewer_WNgx · 2025-11-02

**Soundness:** 3
**Presentation:** 2
**Contribution:** 3
**Rating:** 6
**Confidence:** 3

**Summary:**

The paper focuses on the rank allocation problem in SVD-based LLM compression. Because SVD is applied per linear module, deciding how many singular values to keep under a global compression budget is an important issue. The paper proposes ARA, which learns a monotonic, globally influenced mask over singular values and adds a full-rank guidance loss so the model can decide to keep a layer dense when low-rank is actually worse. Calibration training is done on 256 C4 samples and evaluated on LLaMA2 and Qwen models. The proposed method shows large gains over baseline methods at 80% and 60% compression.

**Strengths:**

* The rank-selection problem itself is important and worth studying.
* The scenario discussed in Section 3.3 where compression actually offers no gain and and the method should fall back to the full-rank weight is also an important but sometimes under discussed topic.
* The paper also shows that the proposed approach is orthogonal to other techniques such as quantization.

**Weaknesses:**

* The method section is not very well presented. I would suggest to improve the clarity for Section 3.2 and consder bringing Algorithm 1 into the main text or explicitly referencing it in the main text.
* The paper keeps saying prior mask methods get stuck in local minima and ARA “reaches the global optimum” (Fig. 2), but what they actually do is add a simple guidance loss term, this does not represent a general optimality result. This statement should consider to be toned down.
* There is no efficiency report in the paper. Since the loss incorporates multiple terms, it would be better to include a wall-clock comparison with baseline methods.

**Questions:**

* The current calibration uses a random subset of the C4 dataset. How robust is this design choice? In particular, how much do the results change if a different subset is sampled or if a different calibration dataset is used?

---

> ### Author Response · Authors · 2025-11-16
>
> Thank you for your valuable comments on our paper. We will further revise the final version based on your suggestions.
>
> **W1:** We will revise the description in Section 3.2, including a more detailed explanation of the design rationale for the guidance loss. We will also change the reference to Appendix A.1 in Section 3 to a direct reference to Algorithm 1.
>
> **W2:** We will change "reaches the global optimum" to "explicitly considers skipping SVD" to make our expression more precise.
>
> **W3:** We report the number of training epochs in Table 5 of the paper. Our method only requires 10 epochs of training on 256*512 tokens. For the Llama 2-7B model, the total time is approximately 20 minutes on a 3090 GPU. Although the loss function includes multiple terms, it converges quickly, and time is not a bottleneck for our method.
>
> **Q1:** Here, we present the final training results under different training datasets (C4, Wikitext2, Alpaca). As shown, different datasets have a certain impact on the results, but compared to uniform compression, the model performance is significantly improved.
>
> **Table 1: Results of Llama 2-7B at 80% compression rate on different training datasets.**
> | Metric | Baseline | Trained on Wikitext2 | Trained on C4 | Trained on Alpaca |
> | :--- | :--- | :--- | :--- | :--- |
> | Wikitext2 PPL | 8.38 | 6.33 | 6.42 | 6.81 |
> | C4 PPL | 20.13 | 10.75 | 10.10 | 11.53 |
> | Acc Avg.(%) | 42.39 | 50.85 | 52.11 | 51.13 |
>
> For different subsets of C4, we used different random seeds for sampling. The results are as follows, which show that different subsets have minimal impact on the experimental results.
>
> **Table 2: Training results of Llama 2-7B at 80% compression rate on the C4 dataset with different random seeds.**
> | Seed | Wikitext2 PPL | Acc Avg.(%) |
> | :--- | :--- | :--- |
> | 3 | 6.42 | 52.11 |
> | 42 | 6.41 | 52.01 |
> | 133 | 6.45 | 52.14 |
> | 233 | 6.47 | 51.87 |

---

### Note · Authors · 2026-01-10

I have read and agree with the venue's withdrawal policy on behalf of myself and my co-authors.